# Use of Acorn Leaves as a Natural Coagulant in a Drinking Water Treatment Plant

**Abderrezzaq Benalia [1], Kerroum Derbal [2], Antonio Panico [3],* and Francesco Pirozzi [4]**

[1]  Laboratory LIPE, Faculty Process Engineering, University of Constantine 3, Ali Mendjeli Nouvelle Ville, 25000 Constantine, Algeria; benalia.abderrezzak@gmail.com

[2]  National Polytechnic School of Constantine, Ali Mendjeli Nouvelle Ville, 25000 Constantine, Algeria; derbal_kerroum@yahoo.fr

[3]  Telematic University Pegaso, 80148 Naples, Italy

[4]  Department of Civil, Architectural and Environmental Engineering, University of Naples Federico II, 80125 Naples, Italy; francesco.pirozzi@unina.it

*  Correspondence: anpanico@unina.it; Tel.: +39-081-768-3434

**Abstract:** In this study, the use of acorn leaves as a natural coagulant to reduce raw water turbidity and globally improve drinking water quality was investigated. The raw water was collected from a drinking water treatment plant located in Mila (Algeria) with an initial turbidity of $13.0 \pm 0.1$ NTU. To obtain acorn leaf powder as a coagulant, the acorn leaves were previously cleaned, washed with tap water, dried, ground and then finely sieved. To improve the coagulant activity and, consequently, the turbidity removal efficiency, the fine powder was also preliminarily treated with different solvents, as follows, in order to extract the coagulant agent: (i) distilled water; (ii) solutions of NaCl (0.25; 0.5 and 1 M); (iii) solutions of NaOH (0.025; 0.05 and 0.1 M); and (iv) solutions of HCl (0.025; 0.05 and 0.1 M). Standard Jar Test assays were conducted to evaluate the performance of the coagulant in the different considered operational conditions. Results of the study indicated that at low turbidity (e.g., $13.0 \pm 0.1$ NTU), the raw acorn leaf powder and those treated with distilled water (DW) were able to decrease the turbidity to $3.69 \pm 0.06$ and $1.97 \pm 0.03$ NTU, respectively. The use of sodium chloride solution (AC-NaCl) at 0.5 M resulted in a high turbidity removal efficiency (91.07%) compared to solutions with different concentrations (0.25 and 1 M). Concerning solutions of sodium hydroxide (AC-NaOH) and hydrogen chloride (AC-HCl), the lowest final turbidities of $1.83 \pm 0.13$ and $0.92 \pm 0.02$ NTU were obtained when the concentrations of the solutions were set at 0.05 and 0.1 M, respectively. Finally, in this study, other water quality parameters, such as total alkalinity hardness, pH, electrical conductivity and organic matters content, were measured to assess the coagulant performance on drinking water treatment.

**Keywords:** acorn leaves; coagulant extraction; drinking water; natural coagulant; solvent; turbidity

## 1. Introduction

Water is a source of life, and in cases of shortage and pollution, it can be a reason for conflict and a source of illness or even death [1,2]. The presence of different impurities in water requires that it be treated to be suitable for specific uses and to improve its quality and ensure health, hygiene and comfort prior to utilization [3–5]. Among all the suitable processes used in drinking water treatment plants, clariflocculation (a sequence of coagulation-flocculation-sedimentation process) is the one of the most common. This process is aimed at reducing the concentration of suspended solids (SS) and non-settling colloidal particles, thus lowering turbidity and improving water quality [6]. The effectiveness of this process is known to be dependent on the nature of the particles, the type and dosage of the coagulant and the pH of the water [7,8].

In aqueous solutions, the dissociation of surface groups, for example hydroxyl groups ($OH^-$), generates electrical charges on the surface of the colloidal particles. These charges add an electrostatic barrier around the particles causing electrostatic stabilization (stabilization based on the presence of surface charges). This stabilizing effect is described in first approximation by the Deryagin–Landau–Verwey–Overbeek (DLVO) theory [9]. The stability of a colloidal system is determined by the sum of the electrical repulsion of the charges on the surface and the Van der Waals attraction. It is a reversible process and simple to implement through modification of the ionic strength (addition of polymers) or the pH of the medium [10,11]. The electric charge around the particles is represented by the zeta potential [10]. In the case of steric stabilization, the particles can be kept at a distance due to the barrier of the organic molecules (e.g., surfactants, polymers, oligomers, etc.) absorbed on the surface of the colloidal particles [12–14].

In several cases, such as in the drinking water treatment process, it is desirable to destabilize the colloidal particles and promote their flocculation rather than stabilize them. In general, drinking water treatment plants use mineral coagulants, such as aluminum sulphate, ferric sulphate, aluminum chloride and ferric chloride, of which aluminum sulphate is the most widely used [15–17]. Apart from the relatively high cost of these mineral-based coagulants, they can be also responsible for detrimental effects on water quality and on public health as a result of their over-dosage. In addition, the residual sludge from these chemical coagulants can be toxic [18,19].

Polymeric materials can be used as flocculants as well. Such polymers act by forming a bridge between the colloidal particles, a process that is called bridging flocculation [9,10,20]. Colloidal particles can flocculate by the bridging mechanism in two ways; the first is by bridging the particles with one polymeric molecule attached to both particles and the second is by bridging the particles through an interaction of polymeric chains attached to different particles.

Furthermore, in order to have low-cost, harmless and environmental friendly surrogate coagulants, in the recent past, several studies have been carried out by testing different natural organic materials to produce bio-coagulants that are as high-performing as the chemical ones. Among these organic materials, cacti [9,20,21], tannins [22], Aloe vera [9], Moringa oleifera [23,24], chestnuts [25] and chitosans [18,26] have shown promising results. Bio-coagulants have been found to be reliable to not only decrease water turbidity [20], but also to remove potential toxic elements (PTEs) [27] and pathogens from water [28]. The advantages of bio-coagulants are biodegradability and non-toxicity.

In this study, we focused our attention on the evaluation of the efficiency of a biodegradable natural product used as a bio-coagulant in the clariflocculation process of raw water. This bio-coagulant was obtained from acorn leaves, which are abundantly available in Algeria, as well as in many other countries worldwide. Several Jar Test assays were conducted on real raw water collected from the drinking water treatment plant located in Mila (Algeria) for the following two main aims: (i) to test the bio-coagulant performance as powder as well as in solution extracted from powder singularly with distilled water and NaCl, NaOH and HCl at concentrations of 0.025, 0.05 and 0.1 M; (ii) to set the optimal concentration of each solvent to reach the highest turbidity removal efficiency. These aspects coupled with tests on raw water actually used to produce tap water represent the novelty of this manuscript.

## 2. Materials and Methods

### 2.1. Analytical Methods

Turbidity was measured using a turbidity meter (HANNA Code: HI 98713, Hanna instruments, Cluj-Napoca, Romania) and expressed in nephelometric turbidity units (NTU). The content of organic matter and total alkalinity hardness were determined according to Standard titrimetric methods [29]. Salinity, electrical conductivity, temperature, pH and total soluble compounds were measured by a multi-parameter instrument (Jenway model 3540, Camlab, Cambridge, United Kingdom).

## 2.2. Raw Water

Raw water was collected from the drinking water treatment plant (36°49′34.89″ N, 6°31′11.81″ E) located in the city of Mila, Algeria. Table 1 shows the main characteristics of this raw water. According to the results shown in Table 1, all the parameters accomplish the Algerian drinking water regulation with the exception of turbidity (13.0 NTU) which is out of the range, being higher than the threshold value (5 NTU).

**Table 1.** Characteristics of used raw water from drinking water treatment plant (Mila, Algeria).

| Parameters | Values | Algerian Standard |
|---|---|---|
| pH | 7.94 ± 0.01 | 6.5–9 |
| Turbidity (NTU) | 13.0 ± 0.1 | 5 |
| Total alkalinity hardness (F°) | 16 ± 1 | 20 |
| Electrical conductivity (μs/cm) | 1244 ± 10 | 2800 |
| Organic matter (mg $O_2$/L) | 2.1 ± 0.2 | 5 |

## 2.3. Preparation of Used Coagulants (Acorn Leaves)

The coagulant was preliminarily obtained as powder from acorn leaves by the following sequence of operations: Acorn leaves were cleaned, washed with tap water, dried at 50 °C, ground and sieved through a 0.35 mm sieve [15,21,30]. The fraction with particles sized lower than 0.35 mm was used in all Jar Test assays. All the steps needed to prepare the bio-coagulant are shown in Figure 1.

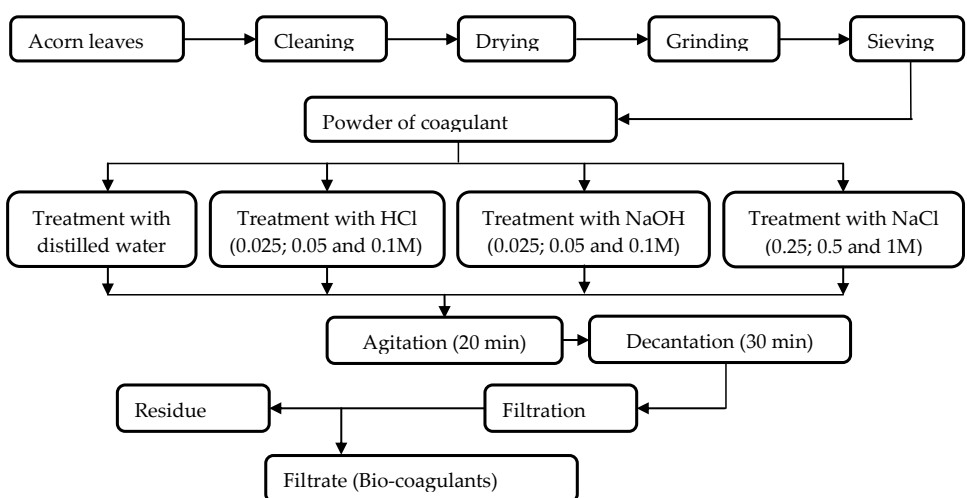

**Figure 1.** Preparation of bio-coagulants.

Furthermore, small amounts of the fine powder (25.00 ± 0.01 g) were added to 1000 ± 10 mL of different solvents. The suspension was stirred using a magnetic stirrer for 20 min to accomplish the extraction of the active coagulating agent from the acorn leaf powder. Solvents used were as follow: (i) distilled water (DW); (ii) solution of NaCl (0.25, 0.5 and 1 M); (iii) solution of NaOH (0.025, 0.05 and 0.1 M); and (iv) solution of HCl (0.025, 0.05 and 0.1 M) [9,21,30]. After 30 min of settling, the supernatant was filtered through a rugged filter paper (0.45 micron) to obtain filtrate extract of the active coagulating agent. The filtrate solutions were used as bio-coagulants in Jar Test assays.

Characterization of Acorn Leaves

The infrared spectrum of acorn leaves was recorded with a Fourier-Transform Infrared Spectrometer. A small amount (1.0 ± 0.1 mg) of acorn leaves was mixed with about 100.0 ± 0.1 mg KBr, and a pellet was prepared using a 10 ton press. The spectrum was recorded in the 4000–500 cm$^{-1}$ range.

Figure 2 shows the infrared spectra of acorn leaves. The strong stretch at $3402.2 \pm 0.1$ cm$^{-1}$ at first sight gives the impression that this was due to the bound OH stretching vibration. The CH group was observed at $2927.7 \pm 0.1$ and $2858.3 \pm 0.1$ cm$^{-1}$. The bands at $2364.6 \pm 0.1$ and $2148.6 \pm 0.1$ cm$^{-1}$ could be representative of C≡N and C≡C respectively. The C=O function of the carboxylic elongation vibration (ester groups) can be assigned to the band at $1743.5 \pm 0.1$ cm$^{-1}$ [31,32].

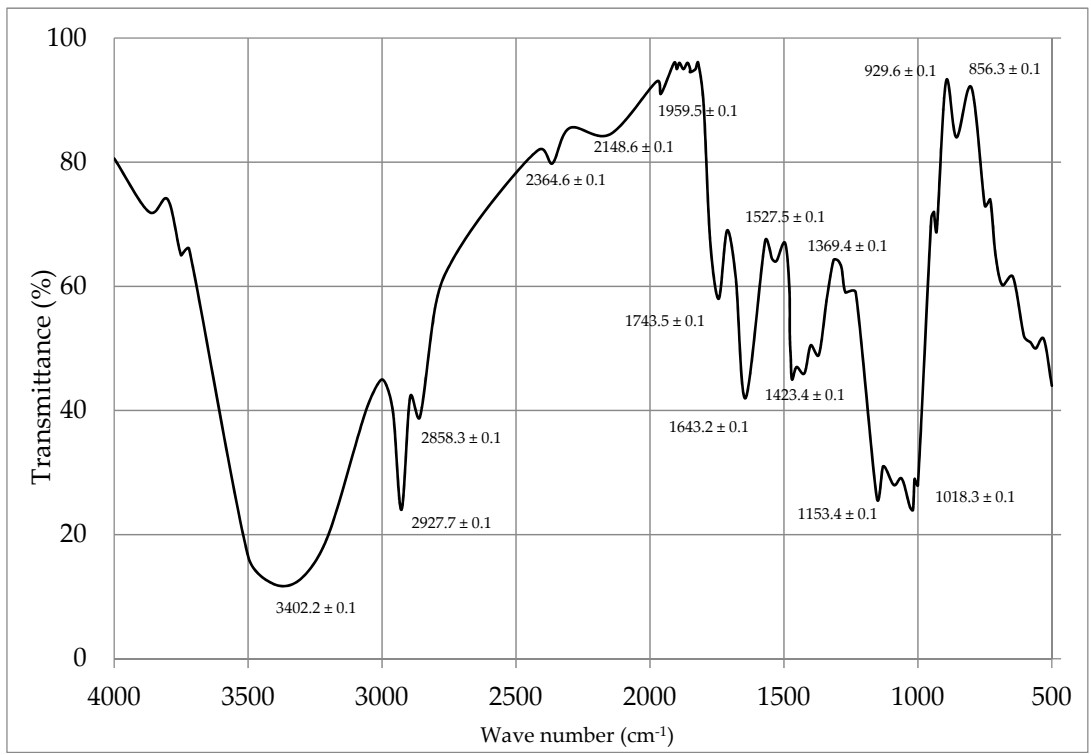

**Figure 2.** Infrared spectra of acorn leaves.

The vibration around $1643.2 \pm 0.1$ cm$^{-1}$ characterizes the C=O bond of the amide, as well as the weak bands at $1959.5 \pm 0.1$ and $1527.5 \pm 0.1$ cm$^{-1}$ confirm the presence of the C=C double bond. Remarkable low transmittances at $1423.4 \pm 0.1$ and $1369.0 \pm 0.1$ cm$^{-1}$ represent the groups $CH_3$ and $CH_2$, respectively. The bands at $1153.4 \pm 0.1$ and $1018.3 \pm 0.1$ cm$^{-1}$ indicate the presence of OH and CO, respectively. The weak band at $856.0 \pm 0.1$ cm$^{-1}$ is supposed to be characteristic of the hydro-peroxides bond [21,33]. The peak formed at $929.6 \pm 0.1$ cm$^{-1}$ confirms the presence of an aromatic group [27].

### 2.4. Experiments (Jar Test Assays)

The Jar Test Apparatus (LI-JTA-125, LABARD, Labard Instruments, Bengali, India) was used to evaluate the performance of the coagulant as powder (AC-powder) and as extracted agent from the four types of solvent, namely distilled water (AC-H$_2$O), solution of NaCl (AC-NaCl), solution of NaOH (AC-NaOH) and solution of HCl (AC-HCl) according to the procedure previously described. To perform the experiments, the 1000 mL beakers were filled with $500 \pm 10$ mL of raw water and placed in the Jar Test Apparatus. During the coagulation step, different concentrations of coagulant were added to each beaker and agitated for 3 min at 160 rpm. During the flocculation step, the mixing velocity was reduced down to 30 rpm for 20 min. Finally, in the settling step, all the suspensions in the beakers were left to naturally decant for 30 min. After that, the supernatant was collected from each beaker to measure the turbidity and other water quality parameters, such as pH, electrical conductivity, organic matter content and total alkalinity hardness. All the Jar Test assays were conducted in

triplicate. The results of turbidity are expressed in percentage point of turbidity removal, using the following formula:

$$\text{turbidity removal (\%)} = \frac{(\text{Initial turbidity} - \text{resitual turbidity}) \times 100}{\text{Initial turbidity}} \qquad (1)$$

## 3. Results and Discussion

### 3.1. Effect of Using of Different Solvents to Extract the Coagulant

In this sub-section, results concerning the effect on the water quality parameters due to the use of the acorn leaves as powder and as coagulating agent extracted with different solvents are presented and discussed.

Effect of Coagulant Dosage on Turbidity Removal Efficiency for Different Solvents

According to international scientific literature, the active coagulant agent extracted from leaves is composed of proteins [23,33,34]. Figure 3a,b shows the effect of the coagulant AC-powder and AC-$H_2O$ on the turbidity removal efficiency: The highest efficiency were 84.77% and 71.6% for AC-$H_2O$ and AC-powder, respectively (see Figure 3a,b). This difference can be explained by considering the higher mobility and accessibility of liquid agent rather than solid to meet the colloids. Figure 3c shows the effects of the coagulant extracted with solutions of NaCl at different concentration (0.25, 0.5 and 1 M) on the turbidity removal efficiency by varying the coagulant dosage. The turbidity removal efficiency was found to increase as the NaCl concentration moved from 0.25 M up to 0.5 M. This result is in agreement with previous research [30] and is related to two phenomena. The first one is known as the "salting-in effect" [30,35,36], in that at higher concentrations of NaCl, more coagulant agent is extracted from the acorn leaves and thus dissolved in the extracting solvent solution [30]. Since the coagulant agent is a protein, when the salt concentration increased, the solubility of the coagulant agent and hence its concentration in the solution also increased. The second phenomenon is the effect of salt (i.e., ionic strength) on particle aggregation [10,37]; in this case, an increase of salt concentration led to an intense particle aggregation due to the compression of the double layer. Moreover, according to Figure 3c, over the concentration of 0.5 M of NaCl, the turbidity removal efficiency decreased. This can be explained by two approaches: The first is the consequence of the "salting-out effect" whereby the solubility of proteins decreased with salt concentration [30,35,36]. The second is the result of the hydration effect occurring when the concentration was significantly high (i.e., 1 M NaCl) [38,39]. The maximum value of removal turbidity efficiency (91.07%) was obtained for the coagulant agent extracted with a 0.5 M NaCl solution.

Figure 3d shows the turbidity removal efficiency obtained with coagulant extracted from acorn leaves with solutions of HCl at different concentration (0.025, 0.05 and 0.1 M). Results highlight an irregular trend as the highest removal efficiency (92.92%) was obtained for coagulant extracted with solution at the highest concentration of HCl (0.1 M) and the lowest efficiency with solution at the mid concentration (0.05 M).

Figure 3e shows the turbidity removal efficiency for coagulants extracted from the acorn leaves using solutions of NaOH (0.025, 0.05 and 0.1 M) in different concentrations. Observing Figure 3e, it can be noted that when the concentration of NaOH in the solution increases, the relating coagulant performs better and the turbidity removal efficiency increases up to reach its maximum (85.92%) with the NaOH solution at 0.05 M. The flection point after 0.05 M in turbidity removal efficiency with a minimum for the coagulant obtained from the 0.1 M NaOH extracting solution can be explained by considering that some proteins may be denatured at this NaOH concentration and hence this phenomenon reduces the protein solubility in the extracting solution [30,40].

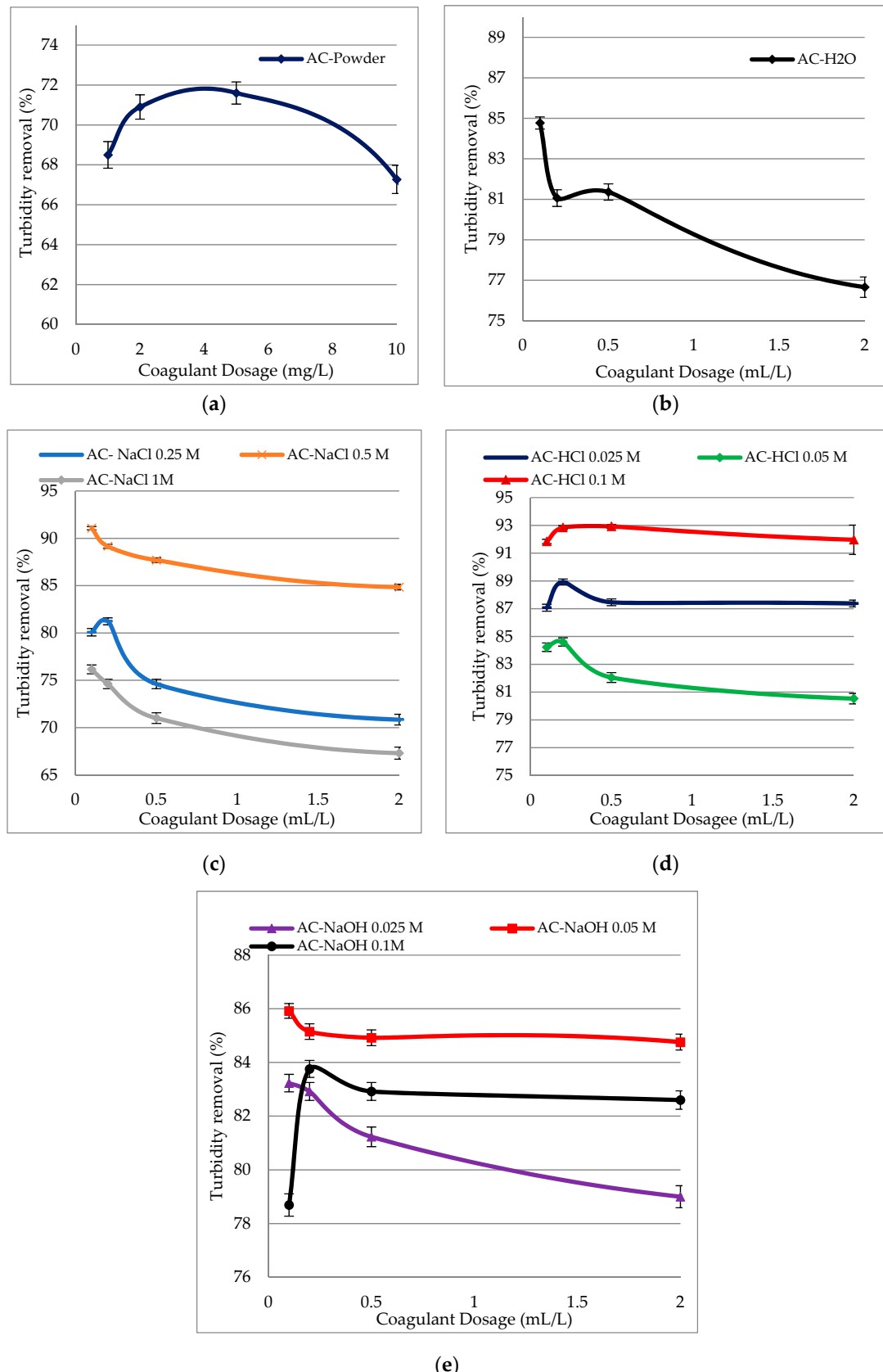

**Figure 3.** Effect of dosage on turbidity removal efficiency for coagulants extracted with different solvents: (**a**) AC powder; (**b**) AC extracted by distilled water; (**c**) AC extracted by NaCl solution; (**d**) AC extracted by HCl solution; (**e**) AC extracted by NaOH solution.

Furthermore, in Figure 3, above the optimum dosage corresponding to the highest turbidity removal efficiency, each further increase in the bio-coagulant dosage caused a decrease in the process performance. This phenomenon can be related to the steric stabilization of particles resulting from an over-dosage of the bio-coagulant [37,41].

### 3.2. Effect of Coagulant Dosage on pH for Different Solvents

Figure 4a–c show that coagulants AC-powder, AC-H$_2$O and AC-NaCl (0.25 M, 0.5 M and 1 M) slightly influence the pH of the raw water and this occurrence can be related to the nature of the considered coagulants. Figure 4d–e confirm that when a solution of HCl is used as a solvent, increasing the dosage of the coagulant, the pH of the raw water decreases to a minimum (7.5), and on the other hand, the pH increases proportionally with the dosage of the coagulant obtained from acorn leaves with NaOH solutions.

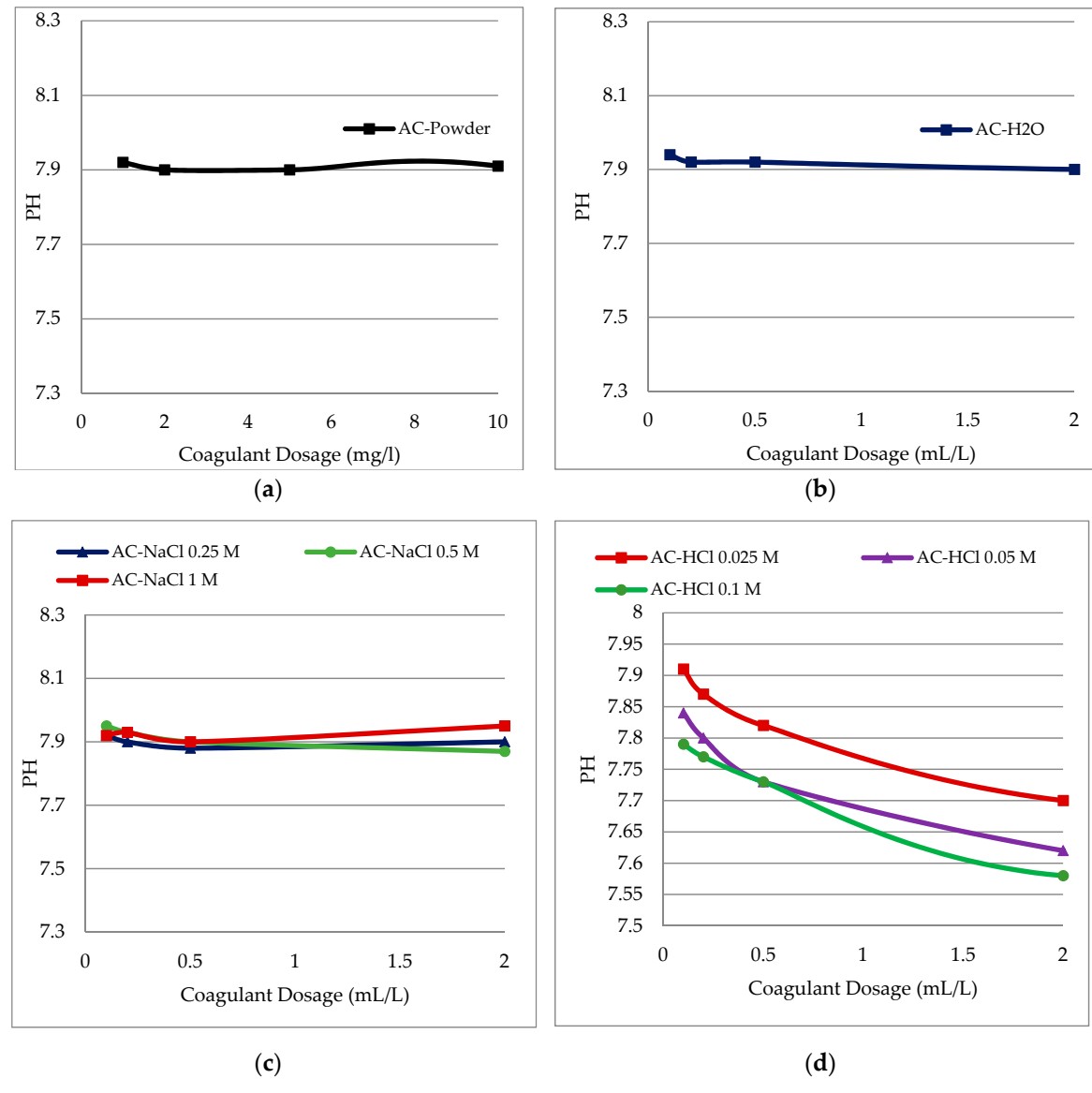

**Figure 4.** *Cont.*

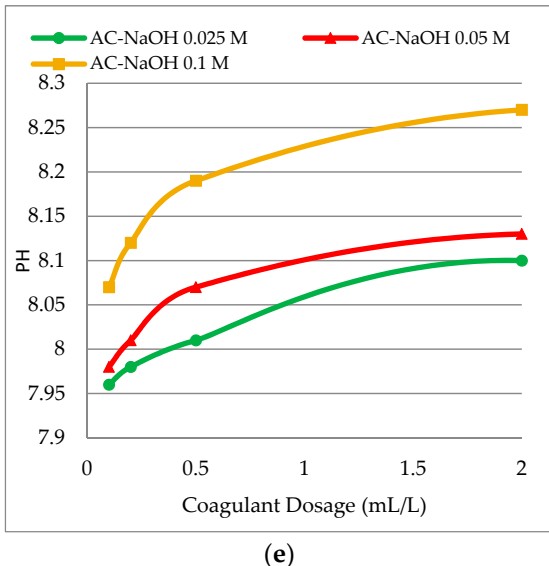

(**e**)

**Figure 4.** Effect of coagulant dosage on pH: (**a**) AC powder; (**b**) AC extracted by distilled water; (**c**) AC extracted by NaCl solution; (**d**) AC extracted by HCl solution; (**e**) AC extracted by NaOH solution.

*3.3. Effect of Coagulant Dosage on Total Alkalinity Hardness, Organic Matter Content and Electrical Conductivity*

In order to see the effect of the use of variously prepared coagulants on the values of some physical-chemical water quality parameters, measurements of the total alkalinity hardness, the organic matter content, as well as the electrical conductivity of the treated water were taken. The values are shown in Table 2. It can be noted that only the values of its parameters corresponding to the optimal dosages of coagulants were considered.

**Table 2.** Characteristics of the treated raw water for the optimal dosage of each coagulant tested.

| Coagulant | Optimal Dosage | Turbidity (NTU) | Total Alkalinity Hardness (F°) | Organic Matter (mg $O_2$/L) | Electrical Conductivity (μs/cm) |
|---|---|---|---|---|---|
| AC-powder (mg/L) | 5.0 | 3.69 ± 0.0 | 16.1 ± 0.2 | 3.8 ± 0.4 | 1244 ± 12 |
| AC-$H_2O$ (mL/L) | 0.1 | 1.97 ± 0.03 | 16.2 ± 0.1 | 2.2 ± 0.2 | 1246 ± 24 |
| AC-NaCl 0.25 M (mL/L) | 0.2 | 2.44 ± 0.07 | 16.2 ± 0.1 | 2.1 ± 0.2 | 1260 ± 16 |
| AC-NaCl 0.5 M (mL/L) | 0.1 | 1.16 ± 0.05 | 16.3 ± 0.2 | 2.0 ± 0.1 | 1260 ± 18 |
| AC-NaCl 1 M (mL/L) | 0.1 | 3.10 ± 0.16 | 16.3 ± 0.3 | 2.4 ± 0.2 | 1269 ± 15 |
| AC-HCl 0.025 M (mL/L) | 0.2 | 1.44 ± 0.03 | 15.4 ± 0.2 | 2.8 ± 0.1 | 1264 ± 16 |
| AC-HCl 0.05 M (mL/L) | 0.2 | 2.00 ± 0.08 | 15.0 ± 0.1 | 2.8 ± 0.5 | 1272 ± 17 |
| AC-HCl 0.1 M (mL/L) | 0.5 | 0.92 ± 0.02 | 13.4 ± 0.1 | 3.0 ± 0.3 | 1280 ± 15 |
| AC-NaOH 0.025 M (mL/L) | 0.1 | 2.18 ± 0.09 | 16.4 ± 0.2 | 1.7 ± 0.3 | 1264 ± 28 |
| AC-NaOH 0.05 M (mL/L) | 0.1 | 1.83 ± 0.13 | 16.6 ± 0.2 | 2.3 ± 0.2 | 1267 ± 16 |
| AC-NaOH 0.1 M (mL/L) | 0.2 | 2.11 ± 0.08 | 17.0 ± 0.1 | 3.0 ± 0.4 | 1274 ± 13 |

Regarding the electrical conductivity, it varies according to the presence of ions, their concentration, their mobility and the nature of the medium. Electrical conductivity is a general indicator of the amount of dissolved solids in the water; the higher the electrical conductivity, the higher the amount of dissolved minerals in the water. Thus, according to Table 2, it should be noted (in case of coagulant extracted with HCl solutions) that increasing the dosage in the extracting solution increases the value of the electrical conductivity. Concerning NaOH, the addition of the coagulant extracted with it increases the value of the electrical conductivity by increasing the concentration of the extracting solution (Table 2).

From Table 2, it is evident that the variation of the total alkalinity hardness is not a function of the added dosage of the various coagulants considered. The table clearly shows that the various coagulants

considered do not have a remarkable effect on the total alkalinity hardness, with the exception of the cases of NaOH and HCl.

When using the NaOH solvent, the increase in total alkalinity hardness is due to the release of $OH^-$, whereas the decrease of total alkalinity hardness with HCl solution is related to the reaction between $OH^-$, $CO_3^{2-}$ and $HCO_3^-$ ions in water and $H^+$ ions released by the HCl solvent.

Regarding the variation of organic matter content, an increase of the values is noticed for all considered coagulants compared to raw water, except for the coagulant obtained by the NaOH 0.025 M solution. This increase is justified by the organic nature of the coagulant [42].

### 3.4. Effect of pH on the Turbidity Removal Efficiency

The parameter pH is very important to enhance the coagulation performance. Figure 5 shows the effect of the pH on the turbidity removal efficiency. According to the results presented in the figure, the optimal pH rage for the highest turbidity removal efficiency is included between 6 and 7.

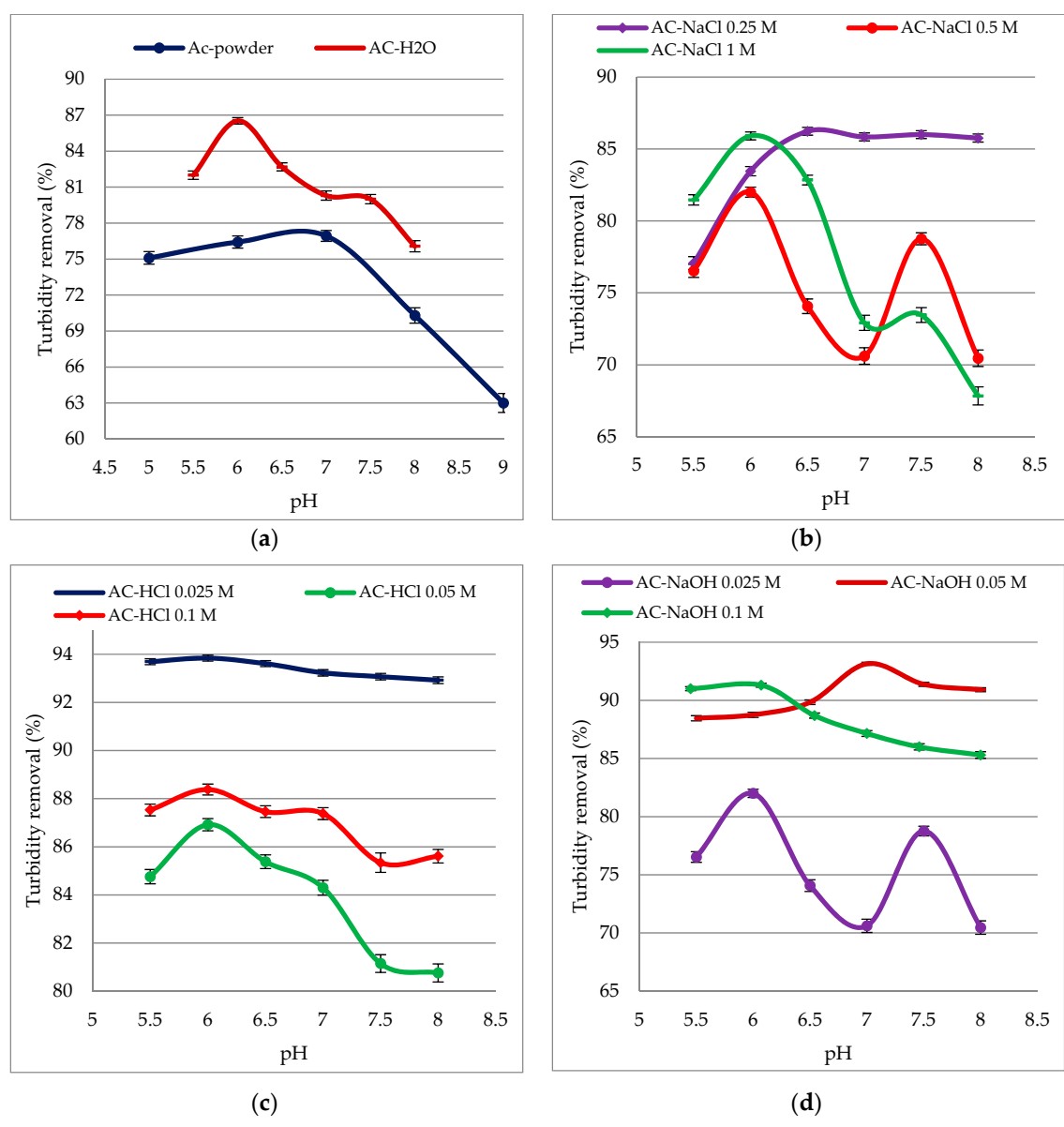

**Figure 5.** Effect of pH on turbidity removal efficiency for different coagulants at their optimal dosage: (**a**) AC extracted by distilled water; (**b**) AC extracted by NaCl solution; (**c**) AC extracted by HCl solution; (**d**) AC extracted by NaOH solution.

## 4. Conclusions

The purpose of this study was to investigate the effectiveness of a natural coagulant obtained and/or extracted from acorn leaves to reduce the turbidity of raw water. This bio-coagulant showed interesting results as powder as much as in solution extracted by distilled water reaching turbidity removal efficiency of 71.6% and 84.77%, respectively.

In order to optimize the use of this material and enhance its efficiency, different solvents in solution at different concentrations were used to extract the coagulating agent from the acorn leaf powder and the results improved remarkably achieving removal efficiency close to 90% for all extracting solutions. In detail: 91.07%, 85.92% and 92.92% respectively for NaCl (0.5 M), NaOH (0.05 M) and HCl (0.1 M). Despite the encouraging results obtained with this material to be used successfully as coagulant in the clariflocculation process, it is relevant to highlight that a high dosage of this material, possibly occurring when the raw water turbidity is high, can worsen the quality of water by increasing the organic matter content, even up to cross the threshold fixed by the national regulation for drinking water.

**Author Contributions:** Conceptualization, K.D. and A.B.; Methodology, K.D., A.P. and F.P.; Formal Analysis, A.B.; Investigation, A.B. and K.D.; Resources, A.B.; Data Curation, K.D., A.P. and F.P.; Writing-Original Draft Preparation, K.D.; Writing-Review & Editing, A.P. and F.P.; Supervision, K.D. and F.P.

**Funding:** This research received no external funding.

**Acknowledgments:** This research has been conducted in the framework of the International Project of Scientific and Technological Cooperation 2016–2018 AL16MO02 promoted by the Ministries of Foreign Affair of Algeria and Italy.

**Conflicts of Interest:** The authors declare no conflict of interest.

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
