# Peer review of "Use of Acorn Leaves as a Natural Coagulant in a Drinking Water Treatment Plant"

_water, doi:10.3390/w11010057_

Round 1

Reviewer 1 Report

This article discusses the effect of a natural coagulant obtained Acorn leaves on the removal of suspended particles. The removal efficiency was evaluated with Jar test. Biodegradable coagulants (macromolecules in this study) are always of great interest to area of water treatment. In this respect, the topic of this study is interesting and the approach is valuable in the aspect of practical application. It is well organized. However, it still requires major revision before being accepted; the details are listed below.  

Major comments:

1. as frequently indicated, the type of the coagulant used in this article seems macromolecules (like proteins), not like inorganic salts. Hence, it is more reasonable to make one paragraph to discuss about the important mechanisms (i.e., polymer bridging, steric stabilization) involved in coagulation/flocculation process when macromolecules-type coagulant/flocculant is used. This will greatly help readers understand the process along with the results in this article. The following articles are the lists that can be referred to and cited for this issue:

(polymer bridging and steric stabilization)

Macromolecule mediated transport and retention of Escherichia coli O157: H7 in saturated porous media, Water research 44 (4), 1082-1093

Surface modification of calcium carbonate with cationic polymer and their dispersibility, Materials transactions 53 (12), 2195-2199

Escherichia coli transport in porous media: Influence of cell strain, solution chemistry, and temperature, Colloids and Surfaces B: Biointerfaces 71 (1), 160-167

(steric stabilization)

Surface characteristics and adhesion behavior of Escherichia coli O157: H7: role of extracellular macromolecules, Biomacromolecules 10 (9), 2556-2564

Contribution of extracellular polymeric substances on representative gram negative and gram positive bacterial deposition in porous media, Environmental science & technology 44 (7), 2393-2399

Escherichia coli O157: H7 transport in saturated porous media: Role of solution chemistry and surface macromolecules, Environmental science & technology 43 (12), 4340-4347

2. section 3.1: linked to the comment 1, there is no discussion on the effect of coagulant dosage on turbidity removal. All data show similar trend at different conditions. Overall, increasing coagulant dosage caused the decrease in turbidity removal. The decreasing trend in the removal efficiency might be due to the steric stabilization of particles with increasing dosage. This should be discussed. The followings are additional references that can be referred to and cited.

Coupled factors influencing the transport and retention of Cryptosporidium parvum oocysts in saturated porous media, water research 44 (4), 1213-1223

Influence of natural organic matter on the deposition kinetics of extracellular polymeric substances (EPS) on silica, Colloids and Surfaces B: Biointerfaces 87 (1), 151-158

3. section 3.1:the removal efficiency initially increased and then decreased with increasing salt (NaCl) concentration. The authors explain the results with salting-in and salting-out effect, which is related to solubility of coagulant used. However, it could be also explained by the effect of salt (i.e. ionic strength) on particle aggregation; increasing salt concentration leads to severe particle aggregation due to the compression of double layer while extreme increase (like 1 M in this article) causes detrimental effect of aggregation due to hydration effect. This potential explanation should be also provided in this article. The following articles are the lists that can be referred to this issue and be cited:

(salt (ionic strength) effect)

Escherichia coli transport in porous media: Influence of cell strain, solution chemistry, and temperature, Colloids and Surfaces B: Biointerfaces 71 (1), 160-167

Influence of natural organic matter on the deposition kinetics of extracellular polymeric substances (EPS) on silica, Colloids and Surfaces B: Biointerfaces 87 (1), 151-158

Influence of nutrient conditions on the transport of bacteria in saturated porous media, Colloids and Surfaces B: Biointerfaces 102, 752-758

(Hydration effect)

Flotation behaviour of malachite in mono-and di-valent salt solutions using sodium oleate as a collector, International Journal of Mineral Processing 146, 38-45

Minor comments:

1. Section 2.3: no reason is given on why the range of NaCl concentration differs from other chemicals (HCl, NaOH).

2. Line 138 and line 144: additional explanation on “Salting-in and salting-out effect” is needed.

Author Response

1. As frequently indicated, the type of the coagulant used in this article seems macromolecules (like proteins), not like inorganic salts. Hence, it is more reasonable to make one paragraph to discuss about the important mechanisms (i.e., polymer bridging, steric stabilization) involved in coagulation/flocculation process when macromolecules-type coagulant/flocculant is used. This will greatly help readers understand the process along with the results in this article. The following articles are the lists that can be referred to and cited for this issue:

We thank the reviewer and accordingly we have added new periods at the introduction section where steric and electrostatic stabilizations have been discussed

Water is a source of life and in case of shortage and pollution, it can be a reason of conflicts and a source of illness and even death, respectively [1, 2]. The presence of different impurities in the water requires treatment to make it suitable for specific uses and to improve its quality and ensure health, hygiene and comfort prior utilization [3, 4, 5]. Among all the suitable processes used in drinking water treatment plants, the clariflocculation (a sequence of coagulation-flocculation-sedimentation process) is the one of the most common. This process is aimed at reducing the concentration of suspended solids (SS) and non-settling colloidal particles, thus lowering the turbidity and consequently improving the water quality [6]. The effectiveness of this process is known to be mainly depending on the nature of the particles, the type and dosage of the coagulant and the pH of the water [7, 8].

In aqueous solutions, the dissociation of surface groups, for example hydroxyl groups (-OH), generates electrical charges on the surface of the colloidal particles. These charges add an electrostatic barrier around the particles causing electrostatic stabilization (stabilization based on the presence of surface charges). This stabilizing effect is described in first approximation by DLVO theory (Deryagin-Landau-Verwey-Overbeek) [9]. The stability of a colloidal system is determined by the sum of the electrical repulsion of the charges on the surface and the Van Der Waals attraction. It is a reversible process and simple to implement through the modification of the ionic strength (addition of polymer) or the pH of the medium [10, 11]. The electric charge around the particles is represented by the zeta potential [10]. In the case of steric stabilization, the particles can thus be kept at a distance thanks to the barrier of the organic molecules (e.g., surfactants, polymers, oligomers, etc.) absorbed on the surface of the colloidal particles [12, 13, 14].

In several cases, such as in drinking water treatment process, it is desirable to destabilize the colloidal particles and promote their flocculation rather than stabilize them. In general, drinking water treatment plants use mineral coagulants such as aluminum sulphate, ferric sulphate, aluminum chloride and ferric chloride, noting that aluminum sulphate is the most widely used [15, 16, 17]. Besides the relatively high cost of these mineral-based coagulants, they can be also responsible for detrimental effects on water quality and on public health in cascade, as result of their over-dosage. In addition, the residual sludge from these chemical coagulants can be toxic [18, 19].

Polymeric materials can be used as flocculants as well. Such polymers act by forming a bridge between the colloidal particles and the process is called bridging flocculation [9, 10, 20]. Colloidal particles can flocculate by the bridging mechanism in two ways: the first way is by bridging the particles with one polymer molecule attached to both particles and the second is bridging the particles through an interaction of polymer chains attached to different particles.

Furthermore, in order to have low-cost, harmless and environmental friendly surrogate coagulants, in the recent past, several studies have been carried out by testing different natural organic materials to produce bio-coagulants as highly performing as the chemical ones. Among these organic materials Cactus [9, 20, 21], Tannin [22], Aloe Vera [9], Moringa Oleifera [23, 24], Chestnut [25] and Chitosane [18, 26] have shown promising results. Bio-coagulants have been found to be reliable not only to decrease water turbidity [20], but also to remove potential toxic elements (PTEs) [27]and pathogens from water [28]. The advantages of bio-coagulants are biodegradability and non toxicity.

In this study, we focused our attention on the evaluation of the efficiency of a biodegradable natural product used as a bio-coagulant in the clariflocculation process of raw water. This bio-coagulant was obtained from Acorn leaves which are abundantly available in Algeria as well as in many other countries worldwide. Several Jar Test assays were conducted on real raw water collected from the drinking water treatment plant located in Mila (Algeria) for the following two main aims: (i) to test the bio-coagulant performance as powder as well as in solution extracted from powder singularly with distilled water and NaCl, NaOH and HCl at the following concentrations 0.025; 0.05 and 0.1M; (ii) to set the optimal concentration of each solvent to reach the highest turbidity removal efficiency. These aspects coupled with tests on raw water really used to produce tap water represent the novelty of this manuscript.

2. Section 3.1:

Linked to the comment 1, there is no discussion on the effect of coagulant dosage on turbidity removal. All data show similar trend at different conditions. Overall, increasing coagulant dosage caused the decrease in turbidity removal. The decreasing trend in the removal efficiency might be due to the steric stabilization of particles with increasing dosage. This should be discussed. The followings are additional references that can be referred to and cited.

We thank the reviewer for this comment and accordingly we have added the following lines

Furthermore in the figures 3, above the optimum dosage corresponding to the highest turbidity removal efficiency, each further increase in the bio-coagulant dosage caused a decrease in the process performance. This phenomenon can be related to the steric stabilization of the particles resulting from an over-dosage of the bio-coagulant [37, 41].

3. Section 3.1:

The removal efficiency initially increased and then decreased with increasing salt (NaCl) concentration. The authors explain the results with salting-in and salting-out effect, which is related to solubility of coagulant used. However, it could be also explained by the effect of salt (i.e. ionic strength) on particle aggregation; increasing salt concentration leads to severe particle aggregation due to the compression of double layer while extreme increase (like 1 M in this article) causes detrimental effect of aggregation due to hydration effect. This potential explanation should be also provided in this article. The following articles are the lists that can be referred to this issue and be cited.

We thank the reviewer for this comment and accordingly we have added new lines to the previous text

Figure 3(c) shows the effects of the coagulant extracted with solutions of NaCl at different concentration (0.25; 0.5 and 1M) on the turbidity removal efficiency by varying the coagulant dosage. The turbidity removal efficiency was found to increase as the NaCl concentration moved from 0.25 M up to 0.5 M. This result is in agreement with a previous research [30] and is related to two phenomena. The first one is known as the “salting-in effect”[30, 35, 36], namely: at higher concentrations of NaCl, more coagulant agent is extracted from the Acorn leaves and thus dissolved in the extracting solvent solution [30]. Since the coagulant agent is a protein, when the salt concentration increased, the solubility of the coagulant agent and hence its concentration in the solution increased. The second phenomenon is the effect of salt (i.e. ionic strength) on particle aggregation [10, 37]: in this case, an increase of salt concentration led to an intense particle aggregation due to the compression of double layer. Moreover, according to figure3(c), over the concentration of 0.5M of NaCl, the turbidity removal efficiency decreased. This can be explained by two approaches: the first is the consequence of the “salting-out effect” whereby the solubility of proteins decreased with salt concentration [30, 35, 36]. The second is the result of the hydration effect occurred when the concentration was significantly high (i.e. 1M NaCl) [38, 39]. The maximum value of removal turbidity efficiency (91.07%) was obtained for the coagulant agent extracted with a 0.5M NaCl solution.

Reviewer 2 Report

1. General Comments:

This paper investigated the "acorn leaves as coagulant in treating water.

Using acorn parts as the coagulant for water treatment have been investigated from beginning of 2000 till now. So authors need to bring up and highlight the novelty of their study.

2. Abstract:

2.1. Page 1, Line 22; "Results of the study indicate ..". Please use PAST SENTENCES. "Results of the study indicated.." 

2.2. Write keywords alphabetically!

3. Introduction:

It is too short! Bring up gap of the knowledge and problem statement. Highlight the novelty of current study.

4. Materials and Methods:

4.1. Write geographic coordinates of the water treatment site.

4.2. Page 2, Table 1: "Conductivity" should be corrected to "Electrical Conductivity"

4.3. Page 2, Line 75: "2.3. Preparation of used coagulants (Acorn leaves)" Support the methodology with some references.

4.4. Quality of Figure 2 is not acceptable! You may re-draw it with some softwares! Not scan it!

5. Results and Discussion:

5.1. Add error bars for Figures 3 and 5

Author Response

1. Using acorn parts as the coagulant for water treatment have been investigated from beginning of 2000 till now. So authors need to bring up and highlight the novelty of their study.

We thank the reviewer for this comment and accordingly we have modified the text where we tried to stress that we have performed in our experiment the application of this bio-coagulant on real raw water and not a synthetic one as commonly done in research previously conducted and furthermore we have tested this material not only as powder (as usually is done), but also as liquid extracted from powder with different extracting solutions at different concentration.

2. Abstract:

2.1. Page 1, Line 22; "Results of the study indicate  ...". Please use PAST SENTENCES. "Results of the study indicated.

We thank the reviewer and accordingly we have corrected the text

2.2. Write keywords alphabetically

We thank the reviewer and accordingly we have corrected the order

3. Introduction

It is too short! Bring up gap of the knowledge and problem statement. Highlight the novelty of current study.

We thank the reviewer and accordingly we have added new periods at the introduction also addressing comments raised by the other reviewers

Water is a source of life and in case of shortage and pollution, it can be a reason of conflicts and a source of illness and even death, respectively [1, 2]. The presence of different impurities in the water requires treatment to make it suitable for specific uses and to improve its quality and ensure health, hygiene and comfort prior utilization [3, 4, 5]. Among all the suitable processes used in drinking water treatment plants, the clariflocculation (a sequence of coagulation-flocculation-sedimentation process) is the one of the most common. This process is aimed at reducing the concentration of suspended solids (SS) and non-settling colloidal particles, thus lowering the turbidity and consequently improving the water quality [6]. The effectiveness of this process is known to be mainly depending on the nature of the particles, the type and dosage of the coagulant and the pH of the water [7, 8].

In aqueous solutions, the dissociation of surface groups, for example hydroxyl groups (-OH), generates electrical charges on the surface of the colloidal particles. These charges add an electrostatic barrier around the particles causing electrostatic stabilization (stabilization based on the presence of surface charges). This stabilizing effect is described in first approximation by DLVO theory (Deryagin-Landau-Verwey-Overbeek) [9]. The stability of a colloidal system is determined by the sum of the electrical repulsion of the charges on the surface and the Van Der Waals attraction. It is a reversible process and simple to implement through the modification of the ionic strength (addition of polymer) or the pH of the medium [10, 11]. The electric charge around the particles is represented by the zeta potential [10]. In the case of steric stabilization, the particles can thus be kept at a distance thanks to the barrier of the organic molecules (e.g., surfactants, polymers, oligomers, etc.) absorbed on the surface of the colloidal particles [12, 13, 14].

In several cases, such as in drinking water treatment process, it is desirable to destabilize the colloidal particles and promote their flocculation rather than stabilize them. In general, drinking water treatment plants use mineral coagulants such as aluminum sulphate, ferric sulphate, aluminum chloride and ferric chloride, noting that aluminum sulphate is the most widely used [15, 16, 17]. Besides the relatively high cost of these mineral-based coagulants, they can be also responsible for detrimental effects on water quality and on public health in cascade, as result of their over-dosage. In addition, the residual sludge from these chemical coagulants can be toxic [18, 19].

Polymeric materials can be used as flocculants as well. Such polymers act by forming a bridge between the colloidal particles and the process is called bridging flocculation [9, 10, 20]. Colloidal particles can flocculate by the bridging mechanism in two ways: the first way is by bridging the particles with one polymer molecule attached to both particles and the second is bridging the particles through an interaction of polymer chains attached to different particles.

Furthermore, in order to have low-cost, harmless and environmental friendly surrogate coagulants, in the recent past, several studies have been carried out by testing different natural organic materials to produce bio-coagulants as highly performing as the chemical ones. Among these organic materials Cactus [9, 20, 21], Tannin [22], Aloe Vera [9], Moringa Oleifera [23, 24], Chestnut [25] and Chitosane [18, 26] have shown promising results. Bio-coagulants have been found to be reliable not only to decrease water turbidity [20], but also to remove potential toxic elements (PTEs)[27]and pathogens from water [28]. The advantages of bio-coagulants are biodegradability and non toxicity.

In this study, we focused our attention on the evaluation of the efficiency of a biodegradable natural product used as a bio-coagulant in the clariflocculation process of raw water. This bio-coagulant was obtained from Acorn leaves which are abundantly available in Algeria as well as in many other countries worldwide. Several Jar Test assays were conducted on real raw water collected from the drinking water treatment plant located in Mila (Algeria) for the following two main aims: (i) to test the bio-coagulant performance as powder as well as in solution extracted from powder singularly with distilled water and NaCl, NaOH and HCl at the following concentrations 0.025; 0.05 and 0.1M; (ii) to set the optimal concentration of each solvent to reach the highest turbidity removal efficiency. These aspects coupled with tests on raw water really used to produce tap water represent the novelty of this manuscript.

4. Materials and Methods:

4.1. Write geographic coordinates of the water treatment site.

We thank the reviewer and accordingly we have provided the geographic coordinates (36° 49’ 34.89” N, 6° 31’ 11.81” E)

4.2. Page 2, Table 1: "Conductivity" should be corrected to "Electrical Conductivity".

We thank the reviewer for this comment and accordingly we have corrected the text

4.3. Page 2, Line 75: "2.3. Preparation of used coagulants (Acorn leaves)" Support the methodology with some references.

We thank the reviewer for this comment and accordingly we have added references

4.4. Quality of Figure 2 is not acceptable! You may re-draw it with some softwares! Not scan it!

We thank the reviewer for this comment and accordingly we have modified the figure

5. Results and Discussion:

5.1. Add error bars for Figures 3 and 5

We thank the reviewer for this   comment and accordingly we have modified the figures

Round 2

Reviewer 1 Report

The authors made great effort on revising their articles, and it has been improved a lot.

I recommend accepting this article as its current form. 

Reviewer 2 Report

It seems that reviewers' comments have been addressed well.